# Are Quality of Randomized Clinical Trials and ESMO-Magnitude of Clinical Benefit Scale Two Sides of the Same Coin, to Grade Recommendations for Drug Approval?

**DOI:** 10.3390/jcm10040746

**Published:** 2021-02-13

**Authors:** Adela Rodriguez, Francis Esposito, Helena Oliveres, Ferran Torres, Joan Maurel

**Affiliations:** 1Department of Medical Oncology, Hospital Clinic of Barcelona,08036 Barcelona, Spain; adrodriguez2@clinic.cat (A.R.); esposito@clinic.cat (F.E.); oliveres@clinic.cat (H.O.); 2Translational Genomics and Targeted Therapeutics in Solid Tumors Group, Institut d’Investigació Biomèdica August Pi i Sunyer (IDIBAPS), 08036 Barcelona, Spain; 3Department of Medicine, University of Barcelona, 08036 Barcelona, Spain; 4Medical Statistics Core Facility, IDIBAPS, Hospital Clinic, 08036 Barcelona, Spain; 5Biostatistics Unit, Faculty of Medicine, Autonomous University of Barcelona, 08036 Barcelona, Spain

**Keywords:** quality randomized studies, ESMO-MCBS, drug approval

## Abstract

The approval of a new drug for cancer treatment by the US Food and Drug Administration (FDA) and the European Medicines Agency (EMA) is based on positive, well-designed randomized phase III clinical trials (RCTs). However, not all of them are analyzed to support the recommendations. For this reason, there are different scales to quantify and evaluate the quality of RCTs and the magnitude of the clinical benefits of new drugs for treating solid tumors. In this review, we discuss the value of the progression-free survival (PFS) as an endpoint in RCTs and the concordance between it and the overall survival (OS) as a measure of the quality of clinical trial designs. We summarize and analyze the different scales to evaluate the clinical benefits of new drugs such as the The American Society of Clinical Oncology value framework (ASCO-VF-NHB16) and European Society for Medical Oncology Magnitude of Clinical Benefit Scale (ESMO-MCBS) and the concordance between them, focusing on metastatic colorectal cancer (mCRC). We propose several definitions that would help to evaluate the quality of RCT, the magnitude of clinical benefit and the appropriate approval of new drugs in oncology.

## 1. How to Measure the Quality of Randomized Clinical Trials

### 1.1. Rating Quality of Evidence and Strength of Recommendations. Is It Time to Change?

The Grading of Recommendations Assessment, Development and Evaluation (GRADE) was published in 2001 by non-oncologic societies [1] and later endorsed by the US Food and Drug Administration. A simplified GRADE adaptation scale [2] offered two grades of recommendation: strong and weak. GRADE has been used by the European Association for the Study of the Liver (EASL) in hepatocellular carcinoma to evaluate clinical recommendations [3]. Other oncologic societies such as ESMO used a different grading system to grade clinical recommendations in metastatic colorectal cancer (mCRC) [4]. Despite it, the levels of evidence in all these grading systems have important weaknesses to adequately interpret the whole-body of evidence in medical oncology. First, terms such as large, randomized trials and with good quality methodology introduce confusion if we do not objectively define the concept of large and good quality methodology. Second, not all randomized clinical trials (RCTs) are analyzed to support recommendations [4]. In addition, different RCTs could have contradictory results, which are tough to analyze. Third, prospective observational studies are ranked below small, randomized trials or large randomized clinical trials with bias suspicion [5] or just missed in other grading classifications [2]. It is important to acknowledge that methodology for causal inference from real-world data has evolved substantially in the last years [6]. Therefore, the strength of evidence should be drawn from a comprehensive literature review and a careful evaluation of the study design, analysis, and interpretation, both in RCTs and in real-world data studies.

### 1.2. Progression-Free Survival Is a Vulnerable Endpoint

Despite the clear definition of events for PFS (Progression Free Survival) (i.e., first of progression or death), the definition to objectively censor patients for PFS is more obscure. Censoring assumes that information after this point is ignorable and, therefore, the risk for censored patients is not different from that for patients still under follow-up. This assumption may be violated in some cases, for instance, when censoring at the time of surgical lung or liver resection or censoring at the time of treatment changes before the progressive disease.

At a more general level, a precise definition of the scientific question pursued by the trial (namely the estimand) is key to designing optimal approaches to handle any intercurrent events (ICE) that might interfere with the endpoint assessment or interpretation [7]. Inappropriate handling of trial ICEs, such as protocol violations or any reason for censoring, will collide with an assessment under a “treatment policy” strategy (i.e., the patient’s status at the end of the study regardless of any ICE). This strategy captures the treatment effect expected in clinical practice after the treatment decision, and it should likely be the main aim in most pivotal trials. Common strategies aligned with a treatment effect following definitions of “while on treatment” (i.e., the net effect observed just, while patients are still on treatment), or even “hypothetical” (i.e., the expected effect should the ICE not occurred) ignore that an ICE actually occurred. Finally, other strategies are possible, but despite several statistical methods that have been proposed for handling ICEs in survival analyses [8], they are also based on assumptions, and it might be difficult to reach a general consensus on the optimal way for handling them.

For these reasons, PFS is a more vulnerable endpoint compared to overall survival, which can easily be retrieved as well as objectively assessed regardless of any ICE under the “treatment policy” strategy. These vulnerabilities include the following.

Censoring in a PFS definition has the effect of selecting the person-time that corresponds to the therapeutic strategy of interest. If the censoring reasons select person-time that corresponds to a strategy that could never be implemented (i.e., a cross-world strategy), such estimation would be useless. For example:

1. Patients who undergo surgical resection of metastases during or after first-line chemotherapy treatment (investigator decision) are usually censored at the time of surgical resection [9]. I.e., the therapeutic strategy that such censoring is defining is “be treated with this experimental drug and never undergo a surgical resection of metastasis, even when indicated”. Because in real life, we would never deny surgical resection of metastasis when indicated, this strategy is not informative to guide therapeutic decisions. A more informative approach is letting patients receive a surgical resection when needed (which can be a downstream effect of the experimental drug) and thus continue their follow-up after such intervention, in the absence of progression.

2. Patients treated with different therapies before the progressive disease (mainly due to severe toxicity) are usually censored the day of the change in the therapeutic strategy. The same principle applies here. Such censoring would correspond to a strategy “be treated with this experimental drug and never change treatment in the absence of progression even if severe toxicity occurs”. Because such a strategy would never be implemented in real life, the results of such analysis cannot guide clinical practice. The more frequent changes in mCRC are due to cetuximab allergy (change to panitumumab) or fluorouracil toxicity after ischemic injury (change to raltitrexed) but censoring these cases would not probably reflect the activity of the new schedule adequately or eventually disease control. We advocate to evaluate these changes as part of predefined therapy and not censor patients until objective disease progression.

If patients were lost to follow-up before the progression of the disease, we recommend that they be censored at the time of the next CT (Computerized Tomography) schedule evaluation. Because events for PFS are defined as the earliest of radiological progression or death, how should we censor a patient that is lost during follow-up without progression disease and died four months after the last follow-up? Should we consider this as an event at the time of death (EMA recommendation), or should it be censored at the time of the last radiological assessment without the progressive disease (FDA recommendation)? We advocate for an intermediate definition and censor it when per-protocol radiological assessments were scheduled before the death (usually CT evaluations range between 6 and 12 weeks).

### 1.3. Concordance between PFS and OS as a Measure of Quality of Clinical Trial Design (QCTD)

Progression-free survival is a relevant clinical endpoint to measure treatment efficacy and has been used as the basis for regulatory approval (FDA and EMA) in many cases. Despite it, the perceived patient value of PFS and the value of PFS per se as the valid surrogate marker is highly discussed in comparison with overall survival and, even with the quality of life [10,11]. Two major arguments have been used for choosing PFS instead of OS as a primary endpoint in RCT in mCRC. First, because the median overall survival of mCRC is two years, first-line therapy needs an extended follow-up compared with PFS, which usually ranges between 8 to 11 months and requires shorter follow-up. Second, because subsequent therapies after disease progression can affect overall survival, PFS will reflect better the activity of first-line therapy [12]. The first reason is a good argument by rapid regulatory agency (FDA and EMA) approval, but it sounds reasonable only in those cases with huge differences in PFS (e.g., pembrolizumab in first-line MSI patients), but not when on the intention to treat analysis HR > 0.65 and differences in median PFS < 3 months. The second reason, although true, is quite debatable, because second-third line therapies or secondary metastatic resections are usually well balanced between arms (between 50 to 70% of patients received second-line therapies, but differences between arms are usually <10%) [13] and probably do not justify huge differences between PFS and OS. In addition, the efficacy (response rate) of second-line therapies in mCRC falls below 20% and PFS range between 4 to 6 months. Therefore, other factors different than second-third line strategies could probably introduce variability between PFS and OS correlation.

We propose two other reasons that potentially can alter the correlation between PFS and OS. First, although large, randomized trials have per definition prognostic characteristics well distributed between arms, pre-planned or non-planned sub-analysis could alter basal patient characteristics that specifically could alter post-progression survival (PPS). Second, targeted agents can alter intrinsic biological characteristics such as consensus molecular subtypes (CMS) at the time of progressive disease [14] and modify clinical (ECOG PS) or tumor biology status (LDH (Lactate dehydrogenase), PAL (alkaline phosphatase)) that potentially can affect PPS. This is of special importance because these characteristics are usually not recorded in RCTs and potentially can influence PFS and OS. For instance, in patients with CMS4 treated with backbone FOLFIRI in FIRE3, the addition of first-line cetuximab instead of bevacizumab with similar PFS (10.5 months with cetuximab and 9.7 with bevacizumab) almost triplicate PPS (29.6 months in cetuximab arm vs. 11.2 in bevacizumab arm). These differences were not observed in CMS2 in FIRE3 or with backbone FOLFOX in CALGB either in CMS2 or CMS4.

Different methods have been used to establish the correlation between PFS and OS. The correlation coefficient (*r*) between PFS (Progression Free Survival) and OS varied between revisions [13,15,16,17,18,19,20]. With chemotherapy alone (without TA) range between *R*^2^ = 0.79–0.82 [15,16] and with chemotherapy plus TA range between *r* = 0.45–0.87 [14,17,18,19]. The slope of the regression line (indicating the% of risk reduction for PFS to estimate the reduction of OS ranged in two studies from 0.54 to 0.67 [15,18]. This means that, for a 10% PFS risk reduction, the OS risk is reduced between 5.4 to 6.7%. Despite this, another study suggests that a 1-month difference in PFS is associated with a 1.3-month difference in OS (slope 1.345). Finally, some authors addressed the correlation between rHR = HR_PFS_ (Hazard Ratio (Progression Free Survival))/HR_OS_ (Hazard Ratio (Overall survival)) [15,20]. When the coefficient is close to 1 or between 0.9 to 1, this would mean that there is little effect on survival related to PFS. When the *r* is less than 0.9 and specially < 0.8, the effect of PFS has more effect on OS. (see Table 1). We must mention that rHR (the ratio of HR_PFS_/HR_OS_) is extremely vulnerable in cases of inadequate control arms (e.g., capecitabine or IFL (irinotecan plus 5-fluorouracil)schedules) when inadequate censures were applied (pe. censoring patients for PFS at the time of metastases resection) or in any subanalysis (either planned or not).

To assess these controversial results, we focused on the slope regression line between PFS and OS and rHR = HR_PFS_/HR_OS_ in 11 studies in first-line therapy comparing doublets with or without bevacizumab [4] and doublets with or without anti-EGFR agents [6] (see Table 2). We also analyzed three additional papers that performed subanalysis in randomized studies with anti-EGFR in the selected group of all RAS WT (Wild Type) [21,22,23,24,25,26,27,28,29,30,31,32,33,34]. We should mention that this analysis supposed a non-preplanned subanalysis of 26% of originally included patients in the OPUS trial, 31% in the CRYSTAL trial and 47% in the PRIME study. In Table 2, we compared trials with non-planned subanalysis with trials with non-planned subanalysis. It seems that the formers adjust the slope regression line between 0.5 and 0.8 and rHR = HR_PFS_/HR_OS_ between 0.75 to 0.90, better than the latter.

## 2. How to Evaluate Clinical Benefit?

The approval of a new drug for cancer treatment by the US FDA and the EMA is based on positive, well-designed randomized phase III clinical trials comparing the investigational treatment with the standard treatment, which theoretically generate unbiased data of efficacy, benefit and safety. Despite it, trials can show statistically significant differences (*p* ≤ 0.05) even when the predefined objectives were not achieved. This could be especially true in trials with larger sample size. Additionally, the value of predefine differences is based on investigator agreements with pharmaceutical companies to obtain regulatory agency approval, and the magnitude of this benefit is not objectively defined. Small clinical benefits with statistical significance compromise global oncology credibility and harm patients to receive treatments based on false expectations. These decisions have private and public economic charges and ethical implications.

A good example of a positive trial design with modest clinical benefit is erlotinib’s approval for pancreatic cancer. The trial design was done to detect a relative risk reduction of 25% (H ≤ 0.75), but the hazard ratio showed a relative risk reduction of 18% (H = 0.82, 95% confidence interval = 0.69 to 0.99), with a statistically significant difference (*p* = 0.038), but a median survival between arms that differed only 10 days. The introduction of new fast-track approval, break-through designations and rapid expansion for new expensive anticancer therapies are increasing. Therefore, the rigor and safety of clinical data supporting FDA and EMA approval are important, especially in the context of a public cost-constrained healthcare system like Europe. This situation justified the development of new tools to objectively assess the magnitude of clinical benefit of anticancer interventions developed by nonprofit organizations such as ASCO and ESMO.

### 2.1. The European Society for Medical Oncology Magnitude of Clinical Benefit Scale (ESMO-MCBS)

The ESMO-MCBS consists of a framework to evaluate the magnitude of clinical benefit for new drugs in the treatment of solid tumors. The first version of the ESMO-MCBS v1.0 was published in May 2015 and a second one, the v1.1, in 2017 [35,36].

This tool is presented in two parts, five forms: Curative setting (Form 1) and palliative setting (Forms 2a, 2b and 3) with two different scales A, B or C and 5, 4, 3, 2, 1, respectively.

This approach incorporates a dual rule:

The observed relative benefit (RB): refers to the lower limit (LL) of the 95% CI (Interval of confidence) for the H compared with a specified threshold value.

The observed absolute benefit (AB) in PFS or OS achieved by the treatment compared with the absolute minimum gain considered as beneficial for the primary endpoint. The second rule is to guarantee that the relevant minimum clinically significant AB is observed.

Congruence between the RB and AB drives the choice of the LL of the 95% CI as a critical statistic improving its sensitivity. Additionally, there is a concordance between the H and the absolute minimum gain in months considered as beneficial according to this ESMO framework. Thus, these rules allow not penalizing treatments whose effects are plausibly congruent with the desired magnitude of RB while penalizing treatments that provide only a trivial observed AB. This approach considers a dual rule: the H refers to the lower extreme of the 95% CI and is used to consider the variability of the estimate, and the observed absolute difference in treatment outcomes (OS/PFS) is compared with the absolute minimum gain considered as beneficial for the primary endpoint. The dual rule approach has been questioned, considering that scoring base on the inferior limit 95% CI of the corresponding H is overly permissive. In fact, it has been demonstrated a decreased phase III RCT that meet the ESMO MBSC criteria, when the estimate H (median) is used instead of the inferior limit 95% CI. This can be explained by the variations in the width of de CI (narrower as the trial has mature data and large sample size and wider as the trial has immature follow-up or/and small trials or any subanalysis). Therefore, trials with immature follow-up or/and smaller sample size or with subanalysis (despite that this subanalysis was pre-planned) can unequivocally increase the wider of the 95% CI interval and decrease the lower limit. Therefore, artificially it would change the punctuation of MCBS.

The correspondence between H and the absolute minimum gain in months considered as beneficial according to this framework is presented by median survival (OS or PFS) for the standard treatment. It can be applied to randomized clinical trials (either with superiority or non-inferiority design). Studies with or without pre-planned subgroup analyses, per definition, would have wider 95% CI and therefore should be penalized. We propose in these cases that estimates instead of a lower range of CI would be taken into account to interpret ESMO-MCBS.

Additional ESMO-MCBS caveats focus on the amount of PFS benefit, especially in trials with < 6 months PFS in the control arm. With the current evaluation form, a benefit of more than 1.5 months and an inferior limit of the H < 0.65 is enough to get maximum punctuation (grade 3). Considering that the correlation between PFS and OS range between 0.5 and 0.77, results in PFS are incongruent with ESMO-MCBS in OS (grade 4 consider a minimum benefit of 3 months in this case).

### 2.2. ASCO Value Framework for Assessing Value in Cancer Care

The American Society of Clinical Oncology (ASCO) value framework (VF) is one of the tools available to evaluate the value of cancer treatments. The ASCO-VF was first published in 2015, and two versions are currently available, the ASCO-VF-NHB (net health benefit) version 2015 (NHB15) and the ASCO-VF-NHB version 2016 (NHB16) [37,38]. The ASCO-VF assigns an NHB16 score with four main components: clinical benefit, toxicity, bonus points (tail of the curve palliation of symptoms, quality of life (QOL), treatment-free interval (TFI)) and drug acquisition (DAC) cost per month. The main critics of the ASCO-VF-NHB16 score argue that it does not consider the designs of the clinical trials and unlikely to represent the general population. They also note that despite it being useful for comparing the clinical impact of a new therapy vs. the control regimen, the ASCO-VF-NHB is not useful to compare between different clinical trials. Finally, some consider it insensitive to palliative care benefits and with unproved physician value.

### 2.3. Concordance between ASCO-VF-NHB16 and ESMO-MCBS

ASCO and the ESMO benefit scores try to discriminate between higher and lower-benefit treatments. ESMO-MCBS assigns a categorical benefit score to positive randomized clinical trials (superiority and non-inferiority trials) for the advanced setting [1,2,3,4] and the curative setting (C, B, A). The ASCO-VF-NHB16 uses a continuous scoring based on randomized (but not necessarily positive) trials, plus one for the advanced setting and another one for the adjuvant setting. ASCO-VF-NHB16 has a larger range of scores, which allow us to better stratify the drugs, but does not provide a therapeutic recommendation. ASCO-VF-NHB16 does not have a threshold of clinical benefit, unlike the ESMO-MCBS framework. The toxicity profile is better and more thoroughly recorded, in the ASCO-VF-NHB 16, although the 5% and 10% incidence cutoffs to assign points could underestimate the high-incidence of toxicities in some drugs. In the ESMO-MCBS, the approach is a simpler grade upgrading or downgrading scheme based on an improvement or deterioration in the quality of life.

The concordance between the two frameworks has been studied, and a globally modest correlation was found. The overall correlation coefficient between the two frameworks in a non-curative setting range between 0.32–0.68 in six different studies [38,39,40,41,42,43]. Concordance between independent researchers was 0.82 (95% CI 0.7–0.9) for ASCO-VF and 0.88 (95% CI 0.8–0.93) for ESMO-MBCS. Absolute concordance is poor, 5% for ASCO-VF and 44% for ESMO-MCBS increasing to 74–80% when deviations within 20 points and 1 grade were allowed.

### 2.4. ASCO-VF and ESMO-MCBS as a Tool to Evaluate Medical Agency Approvals

US Food and Drug Administration (FDA) and the European Medical Agency (EMA) are the final drug payers. Therefore, there is an increasing interest to evaluate how FDA and EMA approvals fulfill ASCO-VF and ESMO-MCBS strict criteria. The different analysis concludes that less than one-third of the drugs approved by the FDA and EMA achieved clinical benefits recommended by both scales [39,44,45,46,47]. (Table 3). Several reasons can justify this discrepancy. First, originally pre-planned differences would not accomplish ASCO-VF and ESMO-MCBS strict requirements. Second, despite that pre-planned differences (usually based on H differences on PFS or OS) were not achieved, significant differences based on the *p* value, was used for approval. It should be noted that any of the previously mentioned reasons probably justify EMA or FDA approval. A reasonable approach would be to state an agreement between regulatory agencies and oncology societies to objectively define optimal ASCO-VF and ESMO-MCBS achievement for drug approval.

## 3. Final Conclusions

We propose several definitions that would help to evaluate the quality of RCT and the magnitude of clinical benefit. These pitfalls and solutions are exposed in Table 4. Ideally, the objective evaluation of both areas would allow establishing an appropriate statement for new drug approval in oncology. Therefore, these considerations ideally should be endorsed by the FDA and EMA or National Health Systems. This specific evaluation has been highlighted and tested in Table 5 in TNBC and esophageal-gastric cancer trials, comparing chemotherapy plus checkpoint blockade vs. chemotherapy alone [48,49,50,51,52,53,54]. Remarkably major differences in ESMO-MCBS evaluation were observed between trials with subgroup analysis based on PD-L1 (Programmed Death-Ligand 1) expression analyzed with Combined positive score (CPS). In brief, we propose drug approval only in the studies that fulfill all the items in both areas (group 1). If studies comply only in part with the required items (group 2), we will propose a conditional drug approval during a reasonably limited time period (for instance, 3–5 years) (Figure 1). In this period of time, a prospective observational (randomized trials are considered not feasible after approval) biomarker-driven study would be expected to be then implemented by the Academia. Sample size in the prospective trial could be designed based on biomarker differences. We believe that this strategy can increase optimal biomarkers discovery for personalized therapy and therefore increase drug efficacy. After the study period, if the prospective observational study is not successful, then drug approval should be denied.

The cost of this prospective observational biomarker-driven study conducted by the Academia would be financed by the pharma, but the cost of the drugs in the study period will be reimbursed by public agencies. During the study period, drugs should only be used in the trial. We propose that at least half of the potential pharma gains would be used to finance the trial.

## Figures and Tables

**Figure 1 jcm-10-00746-f001:**
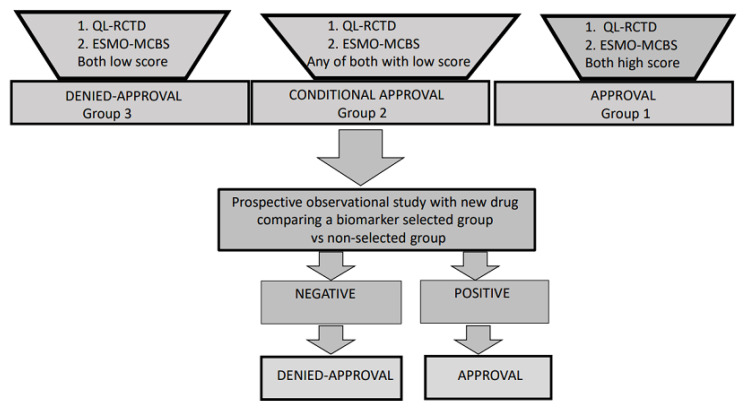
Description of the different types of group for the approval of the drug based on the QL-RCTD (Quality design of randomized controlled trials) and the ESMO-MCBS (European Society for Medical Oncology Magnitude of Clinical Benefit Scale); QL-RCTD—Quality design of randomized controlled trials; ESMO-MCBS—European Society for Medical Oncology Magnitude of Clinical Benefit Scale.

**Table 1 jcm-10-00746-t001:** Correlation coefficient between PFS and OS.

Author	No Trials	Tumor Type	Type of Therapy	Slope Regression Line	rHR	STE
Tang [15]	39	mCRC	CHT	0.54	-	
Buyse [16]	10	mCRC	CHT	0.81	-	0.77
Giessen [17]	50	mCRC	CHT and TA	-	-	
Sidhu [18]	24	mCRC	CHT and TA	0.58 (all) 0.64 (FL)		0.72 to 0.91 (anti-EGFR, WT KRAS subgroup to First Line)
Shi [13]	22	mCRC	CHT and TA	-	-	0.57
Petrell [19]	34	mCRC	CHT and TA	1.34	-	
Tan [20]	51	Across tumor type	TA	-	0.83 (0.79–0.88)	0.50

STE—a surrogate threshold effect; CHT—chemotherapy; TA—targeted agents; rHR—correlation coefficient Hazard Ratio; FL—5-fluorouracil; EGFR-Epidermal Growth Factor Receptor; WT—Wild type. The minimum H effect on the surrogate PFS, which can be translated in the benefit for the HR_OS_ slope of the regression line. Indicates the estimated risk reduction for OS based on estimated risk reduction for PFS. It is desirable that range between 0.50 and 0.81. rHR—the ratio of HR_PFS_/HR_OS_ indicates the strength of the relationship when translating the amount of benefit in PFS to OS. The superior limit of the ratio should ideally be less than 0.9 to guarantee that a benefit in PFS can be translated to a minimum benefit for OS.

**Table 2 jcm-10-00746-t002:** Analysis of PFS and OS correlation in first-line therapy comparing chemotherapy vs. chemotherapy plus anti-EGFR (Epidermal Growth Factor Receptor ) or anti-VEGF (Vascular endothelial growth factor ) agents.

Author	*n* Patients	Treatment Arm	PFS (C vs. E)	OS (C vs. E)	SRL	rHR	HR_PFS_	HR_OS_
Hurwitz [21]	814	IFL +/− BEV	6.2 vs. 10.6	15.6 vs. 20.3	1.06	0.95	0.54	0.66
Saltz [22]	1401	FOLFOX/CAPOX +/− BEV	8 vs. 9.4	19.9 vs. 21.3	1	0.93	0.83	0.89
Guan [27]	214	IFL +/− BEV	4.2 vs. 8.3	13.4 vs. 18.7	1.28	0.71	0.44	0.62
Tebbutt [28]	313	CAP +/− BEV	5.7 vs. 8.5	18.9 vs. 16.4	<0.5	0.61	0.624	0.875
Passardi [29]	376	FOLFOX/FOLFIRI +/− BEV	8.4 vs. 9.6	21.3 vs. 20.8	<0.5	0.76	0.86	1.13
Van Cutsem [25]	1198	FOLFIRI +/− CET	8 vs. 8.9	18.6 vs. 19.9	1.44	0.91	0.68	0.93
Douillard [26]	656	FOLFOX +/− PAN	8 vs. 9.6	19.7 vs. 23.9	2.6	0.96	0.80	0.80
Maughan [32]	729	CAPOX +/− CET	8.6 vs. 8.6	17.9 vs. 17	<0.5	1.1	1.04	0.96
Bokemeyer [24]	337	FOLFOX +/− CET	7.2 vs. 7.2	18 vs. 18.3	<0.5	0.92	0.931	1.015
Tveit [34]	566	FLOX +/− CET	7.9 vs. 8.3	20.4 vs. 19.7	<0.5	0.83	0.89	1.06
Qin [23]	393	FOLFOX +/− CET	7.4 vs. 9.2	17.8 vs. 20.7	1.6	0.91	0.69	0.76
Bokemeyer [33] *	87	FOLFOX +/− CET	5.8 vs. 12	17.8 vs. 19.8	<0.5	0.56	0.53	0.94
Van Cutsem [30] *	367	FOLFIRI +/− CET	8.4 vs. 11.4	20.2 vs. 28.4	2.7	0.81	0.56	0.69
Douillard [31] *	512	FOLFOX +/− PAN	7.9 vs. 10.1	20.2 vs. 25.8	2.6	0.94	0.72	0.77

rHR—the ratio of HR_PFS/_HR_OS_; PFS—progression-free survival; OS—overall survival; C—control arm; E—experimental arm; IFL—irinotecan plus 5-fluorouracil in bolus and leucovorin; BEV—bevacizumab; FOLFOX—oxaliplatin and 5-fluorouracil in the continuous infusion; CAPOX—oxaliplatin plus capecitabine; CET—cetuximab; PAN—panitumumab; FLOX—oxaliplatin and 5-fluorouracil in a bolus; SRL—the slope of the regression line. Indicates the estimated risk reduction for OS based on estimated risk reduction for PFS. r. HR_PFS_/HR_OS._ *studies with non-planned subanalysis.

**Table 3 jcm-10-00746-t003:** Correlation between European Society for Medical Oncology Magnitude of Clinical Benefit Scale (ESMO-MCBS) and ASCO-VF (The American Society of Clinical Oncology value framework) and European Medicines Agency (EMA) and US Food and Drug Administration (FDA) approvals.

Author	No RCTs	Type of Therapy	ESMO-MCBS Benefit% *	ASCO-VF Benefit% *	EMA	FDA
Del Paggio [39]	277	CYT, TA, IT, HT	31	NE	NE	NE
Vivot [45]	51	CYT, TA, IT, HT	25	34	NE	FDA approval
Tibau [46]	105	CYT, TA, IT, HT	38.8 **	NE	NE	FDA approval
Grössmann [47]	70	ND	11 ***	NE	EMA approval	NE

RCTs—randomized clinical trials; CYT—cytotoxic therapy; TA—targeted agents; IT—immunotherapy; HT—hormone therapy; ND—not described; NE—not evaluated; EMA—European Medical Agency; FDA—Food and Drug Agency. * % of drugs approved by EMA or FDA that fulfills ESMO-MCBS or ASCO-VF criteria. ** % ESMO-MCBS analyzed in palliative trials. *** % analyzed with adapted ESMO-MCBS.

**Table 4 jcm-10-00746-t004:** Pitfalls and proposed solutions to evaluate the quality of randomized phase III clinical trials (RCT) (A) and ESMO-MCBS (B).

Type of Analysis	Pitfalls	Solutions
A	1. Missing information of critical prognostic variables at the time of tumor progression	1. Identify in the control and experimental arms critical important variables basally and at the time of tumor progression
A	2. Use inadequate control arm	2. Select adequate control arms
A	3. Modify primary endpoint or use multiple primary endpoints	3. Maintain primary endpoint and use OS as a primary endpoint with an intention to treat analysis
A	4. Plan subgroup analysis as a primary endpoint	4. Plan subgroup analysis as a secondary endpoint mainly to generate a hypothesis
A	5. Miss clear definition of censored patients in the protocol and numbers in the final report	5. Clarify the definition of censored patients in all situations. Specified in the analysis the% of censored patients and the reasons.
A	6. Not evaluate the *r* (HR_PFS_/HR_OS_)	6. Evaluate the *r* and recommend specifically studies that the *r* range between 0.75 and 0.9
A	7. Not evaluate the slope of the curve between PFS and OS	7. Evaluate the slope of the curve between PFS and OS and recommend specifically studies that the slope of the curve range between 0.5 and 0.8
B	8. Consider the inferior limit of 95% CI of H for OS (between 0.70 and 0.75) as an adequate endpoint for ESMO-MCBS punctuation in subgroup analysis	8. If subgroup analysis were done, H estimate (between 0.70 and 0.75) instead of the inferior limit of 95% CI would be recommended to assess ESMO-MCBS
B	9. Not consider the 3 points (% of patients with OS at 2–3–5 years, improvement in H* and median OS) to evaluate the MCBS and do not take the upper punctuation (grade 4 only for OS) to drive positive recommendations for FDA or EMA approvals	9. Consider all 3 points in the MBSC evaluation and take the upper punctuation* (grade 4 only for OS) to drive positive recommendations for FDA or EMA approvals

A: Clinical Trials; B: ESMO-MCBS.

**Table 5 jcm-10-00746-t005:** Controversial results on ESMO-MCBS potentially due to quality design flaws of RCTs (examples with checkpoint blockade inhibitors (CBI) added to conventional chemotherapy in TNBC (Triple Negative Breast Cancer) and esophageal-gastric cancer).

Trial	No. Patient	Treatment Arms	QRCT1 Control Arm	QRCT1 Primary Endpoint *	QRCT1 Endpoint **	QRCT2 SRL	QRCT2 r	ESMO-MCBS PFS	ESMO-MCBS OS	Modified ESMO-MCBS OS ***
**TNBC**										
IMpassion 130 [48]	902	Nab-P + atezolizumab vs. Nab-P	1	0	0	1.6	0.91	1	2	
CPS > 10%	369 (41%)					3	0.92	3	4	4
IMpassion 131 [49]	651	P + atezolizumab vs. P	1	0	0	<0.5	0.78	1	1	
CPS > 1%	292 (44%)					<0.5	073	1	1	1
KEYNOTE-355 [50]	847	CG/P/Nab-P + pembrolizumab vs. CG/P/Nab-P	1	0	0	NA	NA	1	NA	NA
CPS > 10%	323 (38%)					NA	NA	3	NA	NA
**EC/GEJ/G**										
CHECKMATE 649 [51]	1581	FOLFOX/CAPOX + nivolumab vs. FOLFOX/CAPOX	1	0	0	1.3	0.85	1	2	
CPS > 5%	955 (60%)					2.1	0.96	1	4	1
ATTRACTION-4 [52]	724	CAPOX + nivolumab vs. CAPOX	1	1	0	0.13	0.75	2	1	
KEYNOTE-590 [53]	749	CP/FU + pembrolizumab vs. CP/FU#	1	0	0	5.2	0.89	1	3	
CPS > 10%	383 (51%)					2.05	0.82	3	4	2
KEYNOTE-062 [54](CPS > 1%)	507 ****	CP/FU + pembrolizumab vs. CP/FU vs. pembrolizumab	1	1	0	2.8	0.98	1	1	

TNBC—triple-negative breast cancer; EC—esophageal cancer; GEJ—gastroesophageal junction; G—gastric cancer; QRCT; quality of randomized clinical trials; SRL; the slope of the regression line. r.; HR_PFS_/HR_OS_. * if subgroup analysis were planned as a primary endpoint (either for PFS or OS). ** primary endpoint. Unique and based on differences on OS-1 (0 if primary endpoint is multiple or if it includes only PFS). *** Modified ESMO-MCBS evaluate estimates (median of H) instead of the lower limit of 95% CI (applicable only for OS and in subanalysis). **** patients randomized to pembrolizumab plus chemotherapy vs. chemotherapy (the arm with pembrolizumab alone are not evaluated).

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
