# Peer review of "Are Quality of Randomized Clinical Trials and ESMO-Magnitude of Clinical Benefit Scale Two Sides of the Same Coin, to Grade Recommendations for Drug Approval?"

_jcm, 2021, doi:10.3390/jcm10040746_

Round 1
Reviewer 1 Report
Very good setting and a very interesting proposal by the authors. Conclusion sound clear and very well defined. According to my review some revisions needed:
- minor spell check as far as english language is concerned
- Do the authors believe that approval should be given only if primary endpoints are achieved? If not please specify
- "....prospective observational biomarker-driven studies or RCTs should be 3
supported by the pharma, but design would be conducted by Academia...."How do the authors believe that contribution in RCT by pharma and study design by Academia can be combined? In which fashion? - "Group 2 Conditional approval": The authors propose a prospective observational study..In which time frame? which fashion of study? RCT? Which institution should control the quality of this observational study? Do the authors believe that the proposed observational study could include possible bias?
Author Response
4 February 2021
Dear Reviewer,
Thank you for providing the reviewers comments on our manuscript entitled ‘Are Quality of Randomized Clinical Trials and ESMO-Magnitude of Clinical Benefit Scale two sides of the same coin, to grade recommendations for drug approval?’. We were delighted to hear that the editor and reviewer were largely positive and we thank them for helpful and insightful comments. The questions and suggestions made by the reviewer have been addressed and incorporated in the revised version of the manuscript. A point-by-point reply is provided at the end of this letter. We hope that in its present version the manuscript is acceptable for publication in Journal of Clinical Medicine.
We thank you for your time and consideration of our work, and look forward to your future decision.
Yours Sincerely,
Adela Rodríguez, MD
Department of Medical Oncology
Hospital Clinic Barcelona, Translational Genomics and Targeted Therapeutics in Solid Tumors Group, IDIBAPS, University of Barcelona
Barcelona, Spain
Francis Esposito MD
Department of Medical Oncology
Hospital Clinic Barcelona, Translational Genomics and Targeted Therapeutics in Solid Tumors Group, IDIBAPS, University of Barcelona
Barcelona, Spain
Helena Oliveres MD
Department of Medical Oncology
Hospital Clinic Barcelona, Translational Genomics and Targeted Therapeutics in Solid Tumors Group, IDIBAPS, University of Barcelona
Barcelona, Spain
Ferran Torres MD, PhD
Biostatistics Unit
Faculty of Medicine. Autonomous University of Barcelona
Barcelona, Spain
Joan Maurel MD, PhD
Department of Medical Oncology
Hospital Clinic Barcelona, Translational Genomics and Targeted Therapeutics in Solid Tumors Group, IDIBAPS, University of Barcelona
Barcelona, Spain
Comments from Reviewers:
Referee: 1
- minor spell check as far as English language is concerned
In agreement with the reviewer suggestion, we have revised the main spell mistakes.
- Do the authors believe that approval should be given only if primary endpoints are achieved?
We believe that success of the primary endpoint is only one of the items, but that other aspects such as the quality of the trial design and the successful ESMO-MCBS achievement, should also be taken into account, for drug approval. We have clarified this point in the discussion.
- "....prospective observational biomarker-driven studies or RCTs should be supported by the pharma, but design would be conducted by Academia...."How do the authors believe that contribution in RCT by pharma and study design by Academia can be combined? In which fashion?
The cost of this prospective observational biomarker-driven study conducted by the Academia, would be financed by the pharma but the cost of the drugs in the study period, will be re-imbursed by Public Agencies. During the study period, drugs should only be used in the trial. We propose that at least half of the potential gains would be used to finance the trial.
- "Group 2 Conditional approval": The authors propose a prospective observational study..In which time frame? which fashion of study? RCT? Which institution should control the quality of this observational study? Do the authors believe that the proposed observational study could include possible bias?
We would propose a conditional drug approval during a reasonable limited time-period (for instance a 3-5-years). In this period of time a prospective observational (randomised trials are considered not feasible after approval) biomarker-driven study would be expected to be then implemented by the AcademiaSample size in the prospective trial would be designed based on biomarker differences. We believe that this strategy can increase optimal biomarkers discovery for personalize therapy and therefore increase drug efficacy. After the study period, if the prospective observational study is not successful, then drug approval should be denied.
Thank you very much
Adela Rodríguez Hernández
P.D: we have attached the manuscript with the corrections

Reviewer 2 Report
This is an interesting manuscript that discussed the value of the progression free survival as an endpoint in RCTs and the concordance between it and the overall survival as a measure of quality of clinical trials designs. The Authors propose definitions that would help to assess the quality of RCT and the magnitude of clinical benefit. In general, the English language is fine; I suggest to check throughout the text for spelling errors and consistent use of abbreviations. The tables are clear and helpful for the reader. I would also suggest to include further discussion on the potential future direction of this novel approach.
Author Response
4 February 2021
Dear Editor,
Thank you for providing the reviewers comments on our manuscript entitled ‘Are Quality of Randomized Clinical Trials and ESMO-Magnitude of Clinical Benefit Scale two sides of the same coin, to grade recommendations for drug approval?’’. We were delighted to hear that the editor and reviewer were largely positive and we thank them for helpful and insightful comments. The questions and suggestions made by the reviewer have been addressed and incorporated in the revised version of the manuscript. A point-by-point reply is provided at the end of this letter. We hope that in its present version the manuscript is acceptable for publication in Journal of Clinical Medicine.
We thank you for your time and consideration of our work, and look forward to your future decision.
Yours Sincerely,
Adela Rodríguez, MD
Department of Medical Oncology
Hospital Clinic Barcelona, Translational Genomics and Targeted Therapeutics in Solid Tumors Group, IDIBAPS, University of Barcelona
Barcelona, Spain
Francis Esposito MD
Department of Medical Oncology
Hospital Clinic Barcelona, Translational Genomics and Targeted Therapeutics in Solid Tumors Group, IDIBAPS, University of Barcelona
Barcelona, Spain
Helena Oliveres MD
Department of Medical Oncology
Hospital Clinic Barcelona, Translational Genomics and Targeted Therapeutics in Solid Tumors Group, IDIBAPS, University of Barcelona
Barcelona, Spain
Ferran Torres MD, PhD
Biostatistics Unit
Faculty of Medicine. Autonomous University of Barcelona
Barcelona, Spain
Joan Maurel MD, PhD
Department of Medical Oncology
Hospital Clinic Barcelona, Translational Genomics and Targeted Therapeutics in Solid Tumors Group, IDIBAPS, University of Barcelona
Barcelona, Spain
Referee: 2
This is an interesting manuscript that discussed the value of the progression free survival as an endpoint in RCTs and the concordance between it and the overall survival as a measure of quality of clinical trials designs. The Authors propose definitions that would help to assess the quality of RCT and the magnitude of clinical benefit. In general, the English language is fine; I suggest to check throughout the text for spelling errors and consistent use of abbreviations. The tables are clear and helpful for the reader. I would also suggest to include further discussion on the potential future direction of this novel approach
Thank you for the comments. We have reviewed the main spell mistakes and we have clarified in the discussion the future directions of this approach.

Round 2
Reviewer 1 Report
Revision points have been discussed and clarified. No further revisions in the current form of manuscript